# Novel Treatments and Preventative Strategies Against Food-Poisoning Caused by Staphylococcal Species

**DOI:** 10.3390/pathogens10020091

**Published:** 2021-01-20

**Authors:** Álvaro Mourenza, José A. Gil, Luis M. Mateos, Michal Letek

**Affiliations:** 1Departamento de Biología Molecular, Área de Microbiología, Universidad de León, 24071 León, Spain; amouf@unileon.es (Á.M.); jagils@unileon.es (J.A.G.); 2Instituto de Biología Molecular, Genómica y Proteómica (INBIOMIC), Universidad de León, 24071 León, Spain; 3Instituto de Desarrollo Ganadero y Sanidad Animal (INDEGSAL), Universidad de León, 24071 León, Spain

**Keywords:** *Staphylococcus*, enterotoxins, food-poisoning, natural compounds, preventative strategies

## Abstract

Staphylococcal infections are a widespread cause of disease in humans. In particular, *S. aureus* is a major causative agent of infection in clinical medicine. In addition, these bacteria can produce a high number of staphylococcal enterotoxins (SE) that may cause food intoxications. Apart from *S. aureus*, many coagulase-negative *Staphylococcus* spp. could be the source of food contamination. Thus, there is an active research work focused on developing novel preventative interventions based on food supplements to reduce the impact of staphylococcal food poisoning. Interestingly, many plant-derived compounds, such as polyphenols, flavonoids, or terpenoids, show significant antimicrobial activity against staphylococci, and therefore these compounds could be crucial to reduce the incidence of food intoxication in humans. Here, we reviewed the most promising strategies developed to prevent staphylococcal food poisoning.

## 1. Introduction

The bacteria belonging to the genus *Staphylococcus* are generally classified as coagulase-positive or coagulase-negative species. Among all of them, *Staphylococcus aureus* is the leading cause of human disease. Moreover, staphylococcal food poisoning (SFP) is a common disease, and the number of cases had increased continuously since 1884 when the first case was reported to become one of the most common causes of foodborne disease [1]. In addition, coagulase-negative staphylococci and their toxins could also be an important source of food contamination, particularly in ready-to-eat products, milk, cheese, milk chocolate, or canned meat [2,3,4,5,6,7,8,9]. This may lead to food intoxication due to the presence of staphylococcal exotoxins, which were firstly identified in 1992 [7]. Moreover, the inappropriate manipulation of fresh food may lead to outbreaks originated in restaurants because *S. aureus* could also be transmitted from human carriers during food handling [10,11]. Furthermore, food production animals, including pigs, cattle, or chickens, may also be carriers of *S. aureus* [1]. In fact, the use of antibiotics in animal production has increased the incidence of livestock-associated antibiotic-resistant strains [1,12,13]. 

The existence of a robust and straightforward PCR test that detects microorganisms with genes that encode exotoxins has allowed the detection of hundreds of staphylococcal food-poisoning outbreaks every year [14]. However, a high exotoxin production is not directly correlated with a higher disease incidence. Therefore, other alternative analytical methods have been routinely employed to identify the presence of exotoxins in food, including those based on high-performance liquid chromatography (HPLC) and mass spectrometry (MS) [15]. Unfortunately, there is a wide range of exotoxins produced by pathogens from the staphylococcal group, making it challenging to identify the origin of the food intoxication [6]. 

Moreover, the pasteurization of food destroys staphylococci, but it usually does not affect exotoxins’ activity, which may still cause disease in humans after food processing [6]. Besides, some *S. aureus* strains can resist high concentrations of lactic acid, which facilitates their growth in various foods, including cheese, meat, salads, or milk chocolate [1,9,15,16]. 

In addition, coagulase-negative staphylococci in food are important reservoirs of antimicrobial resistance genes, virulence factors, and exotoxins [3,17,18]. This is now considered a significant health problem because these genes could be horizontally transmitted to coagulase-positive staphylococci, which may increase the incidence of foodborne disease [17,19,20,21].

In summary, there is a great interest in developing novel ways of preventing the presence of *Staphylococcus* spp. and their exotoxins in food to reduce the incidence of intoxications. Natural compounds used as food supplements are now considered an up-and-coming strategy to decrease the staphylococcal colonization of food. 

## 2. Staphylococcal Enterotoxins and Virulence Factors

*Staphylococcus* spp. exotoxins present in food are a group of low-molecular-weight pyrogenic proteins of around 22–29 kDa, with important similarities in their secondary and tertiary structures [1,17,22,23,24]. These exotoxins are grouped into three different families depending on their aminoacidic sequence: Staphylococcal enterotoxins (SE), Staphylococcal enterotoxin-like (SEl), and the toxic shock syndrome toxin 1 (TSST-1) [1]. Other toxins related to TSST-1 but showing a different mechanism of action have now been classified as staphylococcal superantigen-like (SSl) [24]. 

Different species of *Staphylococcus* can produce exotoxins, including coagulase-negative strains that are considered non-pathogenic. However, picomolar concentrations of SEs can cause toxic shock syndrome (TSS), fever, hypotension, and multi-organ failure that enhance the disease caused by *S. aureus* [25,26]. In addition, the SE and TSST-1 are also considered superantigens (SAgs) because they have the potential to activate T cells through a complex signaling pathway and stimulate a hyper-inflammatory response [24,27]. SAgs are implicated in the development of sepsis, infective endocarditis, and other complications [28]. 

Among all staphylococcal toxins, SEs have been the most frequently associated with foodborne diseases, causing emesis and T-cell activation [29]. There are at least 26 different SEs characterized, but this number could be even higher [10]. These toxins are resistant to heat and acidity and to the hydrolysis mediated by most proteolytic enzymes [1,17]. 

While SEs and TSST-1 directly activate macrophages and T-cells, SEl and SSl are considered capable of general immunomodulation [24]. However, SEl proteins can induce neither emesis nor T-cell activation [27]. Moreover, some staphylococcal toxins show a proapoptotic activity essential for *S. aureus* colonization [27]. SEs may cause cytotoxicity in intestinal cells, which results in gastroenteritis, vomiting, and gastric inflammation [29]. 

Despite that coagulase-negative strains could be involved in food-poisoning and may even release toxins that cause the lethal toxic shock syndrome [4,22,30], *S. aureus* is still considered the leading cause of staphylococcal gastroenteritis and food intoxication [7,9]. There are important outbreaks caused periodically by *S. aureus* and its toxins, and in most cases, the staphylococcal enterotoxin A (SEA) is involved [5,7]. 

In addition, other virulence factors encoded in the genome of different staphylococcal strains could be putative sources of gastrointestinal diseases. In fact, several virulence factors are essential for the successful colonization of the host, including coagulase, staphylokinase, adhesins, protein A, and *β*-hemolysin [31,32,33]. Their expression is under the control of several regulatory genes and sometimes under the control of noncoding RNAs. The expression of this plethora of virulence factors is indirectly regulated by pH, temperature, and other changing conditions that the pathogen encounters in food or during infection [31,34]. 

Finally, discovering the toxin-antitoxin (TA) system has revolutionized the search for novel therapies against staphylococci [35]. The TA system is composed mainly of two genes encoding an antitoxin (usually located upstream of an operon) and the toxin (located downstream). The first gene is self-regulated, and both the antitoxin and the toxin-antitoxin complex may repress the TA operon’s promoter [36,37]. During infection, the host proteases degrade the antitoxin protein, enabling the expression of the toxin [36]. There is much interest in developing treatments that may inhibit the antitoxin protein’s degradation or even block the antitoxin-toxin interactions, which could be a natural and broad-spectrum antibacterial treatment [36,37].

## 3. Treatments Against Staphylococcal Food Poisoning

Traditionally, the treatments against staphylococcal food-poisoning are focused on either controlling exotoxins or the control of the transmission of the bacteria. The use of antimicrobials to treat staphylococcal food intoxications is not recommended due to the additional release of staphylococcal toxins after bacterial cell death, leading to septic shock. Besides, if antibiotic therapy is administered to patients infected by multidrug-resistant staphylococci, this could facilitate colonization of the gastrointestinal tract once the sensitive intestinal flora is altered. Consequently, antimicrobial-resistant staphylococci may then replicate and secrete more toxins that could aggravate the disease [38]. 

### 3.1. Monoclonal Antibodies and Vaccines

*S. aureus* is a natural commensal of the human skin; however, it can circumvent the host immune system, and it is a facultative intracellular pathogen [39]. These two aspects of the pathogenesis of *S. aureus* are the main causes of the failure of all vaccine candidates tested in humans and based on opsonization [40,41,42]. However, the production of antibodies against staphylococcal extracellular proteins protect patients against sepsis caused by *S. aureus* [43]. Therefore, the development of monoclonal antibodies-based therapies against specific staphylococcal toxins is now considered a very promising strategy to generate protection against *S. aureus* [44]. Importantly, this strategy has been successfully tested in clinical trials against toxins from *Clostridium difficile* or *Escherichia coli* [35]. 

Monoclonal antibodies-based therapies are effective against many other bacteria [45,46], particularly against the most virulent species [46]. However, due to the high number of toxins produced by *S. aureus*, it is becoming clear that therapies based on monoclonal antibodies targeting a single toxin are frequently ineffective [25]. Therefore, there is interest in developing combinatorial therapies against multiple *S. aureus* enterotoxins and other virulence factors, including extracellular or cell-wall anchored proteins [42,45,47,48,49,50,51]. In particular, monoclonal antibodies-based therapies have been developed against multiple staphylococcal enterotoxins and TSST-1 [42,51,52]. Some of these therapies are undergoing clinical trials [45]. This type of therapy could also be used to prevent the spread of the disease in animals infected by livestock-associated multidrug-resistant strains [53]. However, the high cost of this type of treatment makes this strategy difficult to be implemented in animal production. 

In addition, some studies have been focused on developing immunotherapies targeting the staphylococcal α-toxin [42]. This is based on recent evidence showing a high titter of anti-α-toxin antibodies protect against future infections. However, a vaccine developed against different variants of the α-toxin was not effective in humans, despite that it provided effective protection in mouse pneumonia models [54,55]. Nevertheless, these results support the development of multi-target immunotherapies. 

Interestingly, TSST-1 is another important target for the development of immunotherapies. Similar to α-toxin, it has been demonstrated that antibodies generated against TSST-1 may protect patients against future infections [56,57]. Accordingly, 80% of the human population develops antibodies against TSST-1 during the first years of life [58]. A vaccine has been developed to provide immunity against TSST-1 in the remaining 20% of the population, which is also undergoing clinical trials [59,60].

Other interesting targets for immunotherapy-based strategies are virulence factors such as the iron-regulated surface determinants (Isd) proteins, which are located on the extracellular matrix of biofilms produced by *S. aureus* [61]. Moreover, the toxin-antitoxin system could be disrupted by monoclonal antibodies that bind to the toxin but not to the antitoxin [36].

Overall, monoclonal antibodies showed promising results in animal models, but data from clinical trials in humans are still not available or conclusive [48,61]. However, a cocktail of monoclonal antibodies protects mice from *S. aureus* infections very efficiently [48,61]. 

### 3.2. Natural Compounds Against Staphylococcal Infections

As mentioned above, a plethora of toxins, virulence factors, and antimicrobial resistance traits make *S. aureus* a principal cause of foodborne disease. In addition, the biofilm structures created by this pathogen increase its resistance to antimicrobials [62,63]. Fortunately, there is an increasing body of knowledge on natural compounds derived from plants or microorganisms that could be used as dietary supplements to prevent staphylococcal infections and the spreading of their antimicrobial resistance. 

In general, the therapeutic strategies based on natural compounds could be classified by their mode of action in antimicrobial or anti-virulence therapies [64]. Antimicrobial therapies act directly on the bacteria to inhibit their growth, whereas anti-virulence therapies are based on inhibitors of bacterial virulence factors [65]. Importantly, anti-virulence therapies do not directly affect bacterial fitness, and therefore they elicit a limited evolutionary pressure that reduces the development of resistance [35].

Many natural compounds directly inhibit bacterial growth or replication. In particular, polyphenols are very well-known antimicrobial compounds present in significant concentrations in plants (Table 1). Importantly, the combination of polyphenols with other antimicrobial compounds may be synergistic [62], making them very attractive candidates for the development of combinatorial strategies used to prevent staphylococcal food contamination. 

Some of these compounds repress SEs production, whereas others directly interact with enterotoxins and inhibit their mechanism of action. Some of the latter could be used as additives to inhibit SEs-derived food intoxication, particularly in products treated with sterilization or pasteurization processes where staphylococci may be killed. However, their already secreted exotoxins could still be active in contaminated food [1]. Therefore, enterotoxin-directed treatments can be used against *S. aureus*-contaminated and SEs-contaminated food [25], whereas treatments that alter the expression of the genes encoding enterotoxins expression are not effective once SEs are already present in food.

For example, the Muscadine grape’s skin is rich in gallic and ellagic acids, which show significant antimicrobial activity against *S. aureus* [62]. The crude extracts of other plant species such as *Chenopodium album* are rich in phenolic and flavonoid compounds with powerful antibacterial activity against *S. aureus*, equivalent to many antibiotics used in clinical medicine [66]. Aloe vera, black garlic, eucalyptus, or grape seeds could be the source of many other natural compounds with antimicrobial activity [64]. This is due to phenolic and phenolic-derivative compounds and alkaloids, fatty acids, organo-sulfurs, and other aliphatic and cyclic compounds that in total amount to hundreds of molecules with anti-staphylococcal activity [35,64]. 

In addition, some of these plant-derived compounds show activity against SEs [35], which is essential to achieve an all-in-one strategy against staphylococcal food poisoning (Table 1). For example, tomatidine is a well-known antibacterial compound with demonstrated activity against *S. aureus* [67]. Tomatidine is a steroidal alkaloid found in different solanaceous plants, which was first described as a bactericidal agent against small-colony variants of *S. aureus* [68]. However, tomatidine is also a quorum-sensing inhibitor, which alters the expression of many virulence factors, including some toxins [69]. Similarly, the expression of the staphylococcal α-toxin is controlled by allicin, capsaicin, and other amide-derived alkaloids present in chili peppers [35,70], another solanaceous plant. 

Interestingly, anisodamine is an alkaloid produced by a Chinese herb that reduces the TSST-1 concentration in serum and the risk of toxic shock syndrome [71]. Anisodamine is an immunomodulator that inhibits the expression of cytokines such as tumor necrosis factor alpha (TNF-α), interleukin 1 beta (IL-1β), interleukin 8 (IL-8), interleukin 2 (IL-2), and interferon gamma (IFN-γ) expression in a dose-dependent manner, which may eventually reduce the effects of the cytokine storm induced during the toxic shock syndrome [71,72]. 

**Table 1 pathogens-10-00091-t001:** Natural sources of active compounds that could be useful to prevent staphylococcal-food poisoning.

Natural Sources	Active Compounds	Targets	References
Muscadine grape	Gallic and ellagic acids	*S. aureus*	[62]
*Chenopodium album*	Phenolic compounds Flavonoid compounds	*S. aureus*	[66]
Citrus fruits, grapes, and tomatoes	Tomatidine Naringenin	*S. aureus*	[69,73,74]
Fermented orange juice	Naringenin-glycosylated	*S. aureus*	[75]
Garlic	Allicin	*S. aureus*	[70]
Chili peppers	Capsaicin	*S. aureus*	[35]
Chinese herbs	Anisodamine	SEs	[71]
Licorice root	Licochalcone A	*S. aureus*	[76]
Olive oil	4-hydroxytyrosol Tyrosol Vanillic acid p-coumaric acid 4-(acetoxyethyl)-1,2-dihydroxybenzene Pinoresinol 1-acetoxypinoresinol	*Salmonella enterica* *Listeria monocytogenes* *E. coli* *S. aureus*	[77,78,79,80,81,82,83]
Clove oil	Eugenol	SEs	[84]
Wine	Resveratrol Tannins	α-toxin	[35,85]
Mentha	Menthol	SEs	[86]
Hop plant	Xanthohumol	*S. aureus*	[87]
Mustard	Allylisothiocyanate	*S. aureus* *Pseudomonas aeruginosa* *E. coli* *L. monocytogenes*	[88]
Aloe vera	Aloeemodin	*S. aureus*	[64]
Eucalyptus, Mimosa	Pyroligenous acids	*S. aureus* *E. coli* *P. aeruginosa*	[89]
*Caloboletus radicans*	8-deacetylcyclocalopin	*S. aureus*	[90]
*Pleurotus sajor-caju*	*p*-hydroxybenzoic acid *p*-coumaric acid Cinnamic acid	*S. aureus*	[90,91]
Honey	Hydrogen peroxide Gluconic acid Polyphenols	Multiple bacteria *S. aureus*	[92,93,94,95]
Propolis	Polyphenols Waxes Resins Polysaccharides	*E. coli* *S. aureus*	[96,97]
Other natural sources	Cinnamaldehyde Baicalein Apicidin α-cyperone Avellanin C	Quorum sensing *S. aureus* *Bacillus sp.*	[64,98,99]

Moreover, phenolic compounds (in particular, many flavonoids) could be used to control the hemolytic activity and secretion of some SEs (Table 1). For example, licochalcone A may decrease the expression and secretion of SEA and SEB in a dose-dependent manner, and consequently, the release of TNF-α, which ameliorates the adverse effects of these enterotoxins [76]. 

Naringenin is another well-studied natural flavonoid that is present in citrus fruits and tomatoes. Naringenin presents a low antimicrobial activity against *S. aureus*, but it can also inhibit the α-toxin expression at subinhibitory concentrations [73,74]. However, the main handicap of naringenin is its low solubility and, therefore, low oral bioavailability. Nevertheless, the functionalization of the molecule with certain lipophilic groups may enhance its biochemical properties [100]. Furthermore, naringenin-glycosylated forms present in fermented orange juice increase this compound’s absorption profile from the diet [75]. 

Moreover, olive oil possesses many phenolic compounds with important antimicrobial activities, and therefore it is considered a great natural product used for food preservation [77]. The antimicrobial activity of commercial olive oil has been tested against many different bacterial pathogens (Table 1)*,* showing activity against all of them in broth cultures when small quantities of olive oils are added [78]. 

The antimicrobial activity of polyphenols contained in olive oil has been clearly demonstrated [79,80,81]. For example, 4-hydroxytyrosol is a phenolic derivative found in olives that shows SEA-inhibition and bactericidal activity [82]. This compound may be found in plant crude extracts but also in edible olives. However, the polyphenols content can vary depending on the species used, the degree of maturation of its fruits, and the irrigation system used during olive cultivation [79,81]. 

Apart from 4-hydroxytyrosol, the most common phenolic compounds found in olive oil are tyrosol, vanillic acid, p-coumaric, 4-(acetoxyethyl)-1,2-dihydrxybenzene, pinoresinol, and 1-acetoxypinoresinol [77,79,80,81,83]. Due to its complex and variable composition, the olive oil’s polyphenols are collectively named olive oil polyphenol extract (OOPE). 

Other plant-derived polyphenols include eugenol, which is found in clove oil and represses the expression of the genes that encode SEA, SEB, and TSST-1 at subinhibitory concentrations [84]. Wine-derived phenolic compounds such as the stilbenoids (e.g., resveratrol) and tannins showed anti-hemolytic activity [35,85]. Moreover, menthol is a terpene alcohol from plants of the *Mentha* genus that also inhibits the expression of genes encoding exotoxins, specially α-hemolysin, SEA, SEB, and TSST1 [86]. 

Besides, several compounds block biofilm formation or bacterial adhesion to host tissues that are important to reduce the virulence of *S. aureus*. For instance, the hop plant (*Humulus lupulus*) contains xanthohumol, which showed significant antimicrobial activity against *S. aureus* and inhibited its biofilm formation [87]. Moreover, allylisothiocyanate is the product responsible for the pungent taste of mustard, radish, or wasabi, and it is an efficient biofilm inhibitor of many different bacteria, including *S. aureus* [35,88]. There are many more antibiofilm compounds produced by plants, including terpenes, flavonoids, and other phenolic compounds, with variable effectiveness [35,101].

Other plant-derived compounds may exhibit broad anti-staphylococcal activities. For instance, aloeemodin is an inhibitor of the Accessory Gene Regulation C (AgrC) produced by aloe vera, which strongly impacts *S. aureus* because the AgrCA two-component system controls the expression of many virulence factors [64]. On the other hand, eucalyptus and mimosa plants produce pyroligneous acid (PA) with antiseptic activities that have been tested against several bacterial pathogens, including *S. aureus* [89,102]. 

Fungi are another very rich source of antimicrobial compounds that include terpenes, anthraquinones, quinolines, or benzoic acid derivatives (Table 1). For example, members of the genus *Ganoderma* produce many antimicrobial compounds that have been tested against *S. aureus* [90,103]. However, the lack of knowledge on the mechanism of action of these compounds is still impeding their application in the food industry. 

Other European-distributed fungi such as *Caloboletus radicans* produce the anti-staphylococcal compounds 8-deacetylcyclocalopin [90]. In addition, *Pleurotus sajor-caju* produces acid compounds with anti-staphylococcal activity, such as *p*-hydroxybenzoic, *p*-coumaric, and cinnamic acids [90,91]. 

Honey is another well-studied natural product with antimicrobial activity [104]. Honey is composed mainly of sugars, but many other compounds are part of this natural product (Table 1). Firstly, its physiological characteristics, such as its acidity and low water activity, make it a challenging substrate for bacterial growth [92]. However, the complexity of honey composition and its components’ heterogeneity makes it difficult to compare and find the most active compounds with the highest antimicrobial potential [93]. Furthermore, the enzymatic conversion of glucose results in various compounds with antibacterial properties, such as hydrogen peroxide and gluconic acid, which show dose-dependent bactericidal effects [92,93,94,95]. 

Polyphenols are also key antimicrobial molecules present in honey, despite that the polyphenolic profile and its bioactivity changes significantly between many different types of honey [92,105]. Interestingly, the floral source of the honey seems to be a key factor influencing its composition of polyphenolic compounds and their antibacterial activity [106,107]. 

Wax propolis is another bee-derived product made basically of resins, waxes, polyphenols, polysaccharides, volatile materials, and secondary metabolites that show antibacterial, antioxidant, or antiviral activities [96]. Different propolis types have shown activity against different Gram-positive and Gram-negative bacteria, including *S. aureus* [97]. Similar to honey, the floral origin of the propolis determines its composition and antimicrobial activity.

Finally, many natural products inhibit the AgrC-based quorum sensing, biofilm formation, and cell-to-cell communication in *S. aureus* [64,65,108,109]. These compounds are frequently produced by fungi or plants and include cinnamaldehyde, baicalein, apicidin, α-cyperone, or avellanin C [64,98,99]. In addition, some bacteria with probiotic potential, such as *Bacillus* spp., may also produce very effective quorum-sensing inhibitors [110]. 

## 4. Conclusions

The incidence of antimicrobial-resistant bacterial infections is increasing worldwide, and the development of new antimicrobial therapies is not keeping pace with the acquisition and transmission of antibacterial resistance. *S. aureus* is one of the most important human pathogens, and it is quickly acquiring resistance to last-resort drugs used in clinical medicine. Moreover, staphylococci and their exotoxins are important sources of food contamination. There are many promising preventative and therapeutic strategies against staphylococcal food intoxications, but very few have been tested in vivo, and a limited number of clinical trials have been conducted with these compounds. 

One of these exceptions is the flavonoid naringenin, which has been recently tested in clinical trials [111]. However, naringenin has a low oral bioavailability; thus, new naringenin-glycosylated derivatives are currently developed to improve its absorption profile. Many other natural compounds with antimicrobial activity against staphylococci have only been tested in preclinical trials due to their low absorption, distribution, metabolism, or excretion properties. 

Nevertheless, the natural products with antimicrobial activity against staphylococci have the potential to be used as food additives alone or in combination to prevent food-poisonings. However, more research is required to test the dosage and stability of the compounds with the best antimicrobial profiles. 

Plant-derived polyphenols are one of the most important sources of antimicrobial compounds with activity against *S. aureus*. Flavonoids, terpenoids, and other important antimicrobial compounds could be found in citrus fruits, grapes, honey, garlic, and other inexpensive food that undoubtedly may impact the incidence of staphylococcal food intoxications and the spread of antimicrobial resistance. Combining different natural compounds could enhance their antimicrobial or antitoxin activities, but more research is needed to evaluate their possible synergistic effects.

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
