# Peer review of "Novel Treatments and Preventative Strategies Against Food-Poisoning Caused by Staphylococcal Species"

_pathogens, 2021, doi:10.3390/pathogens10020091_

Round 1

Reviewer 1 Report

In this review, authors described natural active compounds as probable treatment and preventive strategies against Staphylococcal food poisoning. The manuscript was well-written, and I have some comments as follows:

  1. What are the inclusion criteria of references used in this review?

  1. In conclusion section, please mention the possible limitations in using those natural active compounds to prevent Staphylococcal food poisoning.

  1. How do authors think about the dosage and stability of those natural compounds for effectiveness in treatment and prevention of Staphylococcal food poisoning?

Author Response

Many thanks for your comments. Regarding the inclusion criteria, we would like to emphasize that this is not a systematic review of the literature. Our main aim was to produce a current but an amenable revision of the literature focused on very promising strategies to control staphylococcal food poisonings. The general objective was to provide the reader with a general introduction to many strategies based on immunotherapy or natural compounds. Regarding the latter, we have focused on natural products that may be obtained from inexpensive sources, and that could be used as food supplements to prevent staphylococcal food poisonings.

Regarding the limitations in using natural active compounds to prevent staphylococcal food poisonings, one of these limitations is in fact the dosage and stability of these compounds to become effective as diet supplements. To address this point, we have expanded the conclusions to explicitly mention that many of these compounds have low absorption, distribution, metabolism, or excretion properties and that further research is required to improve these natural products. In this line, we have also expanded on naringenin as a promising example of a natural compound with low oral bioavailability that could be improved by producing naringenin-glycosylated derivatives. Therefore, our final message is that these preventative or therapeutic strategies are very promising, but further research is needed. We hope that we have addressed all your concerns by expanding our revision in this manner.

Reviewer 2 Report

The manuscript gives information on gastrointestinal disease and food-poisoning caused by Staphylococcal enterotoxins. An other report provides up-to-date brief information on Staphylococcal enterotoxns and virulence factors.

However, the authors list the references irrespective of the fact whether they come from clinical studies and/or non-clinical/experimental studies based on statistically relevant data; from anecdotal clinical or experimental evidence or of treatment attempts.

The most important suggestion to the authors would be: to categorize the references according to these or equivalent criteria or if they would mention this situation in an additional paragraph. (e.g. listing the "natural sources of active compounds that could be useful to prevent Staphylococcal food poisoning" as suggested treatment attempts.)

Author Response

Many thanks for this suggestion. Unfortunately, very few of the compounds listed here have reached the clinical testing phase due to their low absorption, distribution, metabolism, or excretion properties, which is now mentioned in the conclusions. However, we have also included in the conclusions a very illustrative example based on naringenin. This natural compound could be effective against staphylococcal food poisonings, but its oral bioavailability is low. Nevertheless, naringenin bioavailability could be improved by producing glycosylated derivatives of this flavonoid. Therefore, our final message is that these strategies are very promising, but their full potential has not been fully unravelled yet. We hope that we have addressed your concerns by expanding our revision in this way.

Round 2

Reviewer 2 Report

Dear Authors,

Thank you for accepting and responding to the criticism.